# Markers of Oxidative Stress in Obstetrics and Gynaecology—A Systematic Literature Review

**DOI:** 10.3390/antiox11081477

**Published:** 2022-07-28

**Authors:** Michalina Anna Drejza, Katarzyna Rylewicz, Ewa Majcherek, Katarzyna Gross-Tyrkin, Małgorzata Mizgier, Katarzyna Plagens-Rotman, Małgorzata Wójcik, Katarzyna Panecka-Mysza, Magdalena Pisarska-Krawczyk, Witold Kędzia, Grażyna Jarząbek-Bielecka

**Affiliations:** 1Specialty Trainee in Obstetrics and Gynaecology, Princess Alexandra Hospital NHS Trust, Harlow CM20 1QX, UK; 2Medical University of Warsaw, 02-091 Warsaw, Poland; kate.rylewicz@gmail.com; 3Poznan University of Medical Sciences, 61-701 Poznań, Poland; ewaa.majcherek@gmail.com; 4INVICTA Fertility and Reproductive Clinic, 80-850 Gdansk, Poland; katarzyna.grosstyrkin@gmail.com; 5Dietetic Department, Faculty of Physical Culture in Gorzów Wielkopolski, Poznań University of Physical Education, 61-871 Poznań, Poland; m.mizgier@diaeteticus.pl; 6Institute of Health Sciences, Hipolit Cegielski State University of Applied Sciences, 62-200 Gniezno, Poland; plagens.rotman@gmail.com; 7Department of Physiotherapy, Faculty of Physical Culture in Gorzów Wielkopolski, Poznań University of Physical Education, 61-701 Poznań, Poland; malgo_wojcik@interia.pl; 8Department of Perinatology and Gynaecology, Poznan University of Medical Sciences, 61-701 Poznań, Poland; katarzyna.panecka@interia.pl (K.P.-M.); witold.kedzia@poczta.fm (W.K.); grajarz@o2.pl (G.J.-B.); 9The President Stanislaw Wojciechowski Calisia University, 62-800 Kalisz, Poland; magmp@op.pl

**Keywords:** pregnancy, oxidative stress, reproduction, fertility, antioxidants, metabolism

## Abstract

Oxidative stress has been implicated in many diseases, including reproductive and pregnancy disorders, from subfertility to maternal vascular disease or preterm labour. There is, however, discrepancy within the standardized markers of oxidative stress in obstetrics and gynaecology in clinical studies. This review aims to present the scope of markers used between 2012 and 2022 to describe oxidative stress with regard to reproduction, pregnancy, and pregnancy-related issues. Despite the abundance of evidence, there is no consensus on the set of standardised markers of oxidative stress which poses a challenge to achieve universal consensus in order to appropriately triangulate the results.

## 1. Introduction

Oxidative stress (OS) is defined as a state of imbalance between pro-oxidant molecules, including reactive oxygen and nitrogen species, and antioxidant defenses. ROS (reactive oxygen species) and RNS (reactive nitrogen species) have a significant role in human bodies’ oxidative balance. Those molecules are recognised as important factors in redox signaling, growth regulation and initiating, mediating, or regulating the cellular and biochemical complexity of oxidative stress [1]. Lack of balance in that field can cause serious implications, such as oxidative damage and tissue dysfunction [2]. That process leads to various consequences for the organism such as cancer [3], heart disorders, cardiovascular disease, atherosclerosis, hypertension, reperfusion injury, diabetes mellitus, or neurodegenerative diseases [4]. Furthermore, it can especially affect pregnant patients as ROS and RNS are identified as factors causing preeclampsia, placental diseases, and premature birth [5].

The excess of reactive oxygen species can lead to cellular damage of lipids, DNA, and proteins. The consequence of disturbed haemostasis is also the damage of mitochondrial and nuclear DNA as well as lipid peroxidation. Unsaturated fatty acids and other lipids undergo oxidation by becoming peroxides. These compounds, such as MDA (malondialdehyde), impair functioning cells through disorders of structure and breaking cell membranes and also changing functions of receptors. Total antioxidant status (TAS) can determine quantitatively the influence of oxidative stress in a human body and degree of protection against its activity. TAS is a parameter coming from evaluation of blood plasma that finds expression mainly in a number of thiol groups, proteins of blood plasma, and concentration of uric acid [6].

The aim of antioxidants is to protect cells from damage and support, maintaining the integrity of the cell membrane as well as peroxidation reactions. Most commonly used antioxidants—such as vitamins (A, E, C) and elements such as zinc, iron or selenium—have potential protective functions for disease prevention. However, despite overwhelming evidence that the oxidative stress affects reproduction and pregnancy, there is so far limited evidence that antioxidants supplementation is significant with regard to its effects on combating oxidative stress or reversing pathological processes. Some studies suggest the positive effect of antioxidants such as N-acetylcysteine [7], vitamins C and E, L-arginine, and resveratrol on pregnancy-related medical conditions such as preeclampsia [8], intrauterine growth restriction, as well as on pregnancy outcomes in women with polycystic ovarian syndrome [9]. Nonetheless, further studies are needed to draw any conclusions regarding the aforementioned antioxidants’ effectiveness as the currently available data are insufficient [10,11].

The lack of balance between pro-oxidant and antioxidant agents might cause multiple negative reproductive health outcomes, such as polycystic ovary syndrome (PCOS), subfertility, or endometriosis. Pregnancy complications—such as miscarriages, gestational diabetes and preeclampsia, fetal growth restriction, and preterm labour—can also develop in response to oxidative stress. Studies have shown that both being underweight and overweight—as well as certain risk behaviors such as recreational alcohol use, smoking, or illicit drug use—can increase production of excess free radicals, which has a known effect on reproductive and perinatal health. Moreover, being exposed to pollution in the environment or known “endocrine disruptors” present in domestic products can lead to imbalance towards pro-oxidative stress and contribute to struggles with fertility [12].

There have been multiple attempts to define oxidative stress [13,14,15,16,17,18]. Costantini [13] in his commentary proposes biochemical and biological definitions of oxidative stress. Some of the definitions focus on the damage created at the biochemical level and imbalance towards pro-oxidants causing stress at the cellular level [14]; Other definitions look into the biomolecular damage caused by reactive species attacking the constituents of living organisms [15,16]. However, biochemical definitions of oxidative stress can also focus on the effects on cellular signaling and its disruptions [17,18]. Moreover, many authors are not only using different approaches to the definition of oxidative stress but also different parameters to assess oxidative stress. There is no unity in tests and markers—some assess reactive oxygen species (ROS), TAC, antioxidants potentials, or even inflammatory markers as proxies of oxidative stress. Given this discrepancy, our research team decided to look into the definitions and the oxidative stress markers used in literature with regard to obstetrics and gynaecology.

## 2. Materials and Methods

Two independent reviewers have searched medical and public databases—including Cochrane, PubMed, Google Scholar, and MEDLINE—using the search terms and MeSH terms such as: “oxidative stress”, “antioxidant*”, “pregnancy”, “gyn(a)ecology”, “obstetrics”, “reproduction”, and “fertility”. We were searching for papers which presented the parameters used to describe oxidative stress and its markers and discussed female reproductive tract disorders, subfertility as well as pregnancy and pregnancy-related issues.

The inclusion criterion was for the paper to be published in the peer-reviewed journal in the last 10 years (2012–2022). No limitation to language of the publication or type of the study were made. Papers discussing male infertility and reproductive issues were excluded.

The papers were then vetted by the review team against inclusion criteria and the final list of papers was presented in a table looking at population, materials used to assess oxidative stress, parameters assessed, which reproductive or pregnancy-related issue, which intervention (if any) was introduced, and what the outcomes were of each study.

## 3. Results

### 3.1. Study Characteristics

The team of reviewers have identified 46,436 records, 600 of which were then screened. Then, 105 were retrieved and assessed for eligibility and ultimately 83 papers were included into final review. Two reviewers independently screened databases, assessed against the inclusion criteria and eligibility.

Different types of studies were included in the analysis: 45 case-control studies, 24 randomized controlled clinical trials, 9 cohort studies, and 5 cross-sectional studies.

The process is illustrated in Figure 1 below. The list and paper characteristics are included in Appendix A, Table A1 at the end of the manuscript.

### 3.2. Markers of Oxidative Stress

We found that a plethora of different markers of oxidative stress were used. This includes malondialdehyde (MDA), nitrous oxide (NO), reactive oxygen species (ROS), total antioxidant capacity (TAC), total antioxidant activity (TAA), superoxide dismutase (SOD), glutathione peroxidase (GPx), glutathione peroxidase (4 GPx), glutathione reductase (GR), lipid peroxidation (LPO), 8-hydroxydeoxyguanosine (8-OHdG), oxidised glutathione (GSSG), catalase (CAT), superoxide (O_2_^−^), Paraoxonase (PON-1), oxidative stress index (OSI), hs-CRP, 8-iso-prostaglandin F2α (8-iso-PGF2α), prostaglandin F2α (PGF2α), gluthatione (GSH), and glutathione transferase (GST).

### 3.3. Materials

Materials used for examination of the markers are characterized by high diversity. Researchers used mostly blood (serum or plasma) (*n* = 68), placenta (*n* = 8), urine (*n* = 6), Wharton’s jelly mesenchymal stem cells from umbilical cord (*n* = 1), or saliva (*n* = 4). Ovarian follicular fluid (*n* = 9), peritoneal fluid (*n* = 2), and granulosa cells (*n* = 3) were used when examining reproductive health issues such as polycystic ovarian syndrome and endometriosis.

### 3.4. Pregnancy-Related Conditions

The team divided emerging themes into pregnancy related and reproduction related conditions. Among pregnancy related conditions, the team distinguished pre-eclampsia, gestational diabetes mellitus, preterm birth, as well as issues with regard to general antenatal care such as association with birth weight or iron supplementation. Neonatal outcomes were not analyzed for the purpose of this study.

#### 3.4.1. Pre-Eclampsia

We retrieved 10 articles about the role of oxidative stress in pre-eclampsia. In total, 17 biomarkers of OS were measured with the number of studies that they were identified in put in brackets (*n* = X): MDA (*n* = 5), TAS (*n* = 4), GSH (*n* = 3), CAT (*n* = 2), TOS (*n* = 2), GSSG (*n* = 1), TAC (*n* = 1), OSI (*n* = 1), SOD (*n* = 1), GPx (*n* = 1), NO (*n* = 1), carbonic anhydrase IX (*n* = 1), peroxynitrite (ONOO^−^) (*n* = 1), paraoxonase (PON-1) (*n* = 1), O_2_^−^ (*n* = 1), 8-OHdG (*n* = 1), and 8-isoprostane (*n* = 1) [11,12,13,14,15,16,17,18,19,20].

#### 3.4.2. Gestational Diabetes Mellitus (GDM)

There is great diversity of markers in papers researching correlation between OS and GDM. In 30 studies, 43 biomarkers were measured. The markers that were most frequently measured were: MDA (*n* = 17), TAC (*n* = 12), GSH (*n* = 9), GPx (*n* = 6), SOD (*n* = 6), CAT (*n* = 4), NO (*n* = 4), and 8-isoprostane (*n* = 4).

The rest of parameters were oxidative stress index-OSI (*n* = 3), GST (*n* = 2), GR (*n* = 2), uric acid (*n* = 2), xanthine oxidase (*n* = 2), TOS (*n* = 1), TNF-α (*n* = 1), IL-10 (*n* = 1), paraoxonase (PON-1) (*n* = 1), inactivation of aldehyde dehydrogenase (*n* = 1), irisin (*n* = 1), bilirubin (*n* = 1), 8-OHdG (*n* = 1), sulfhydryl groups (*n* = 1), plasma and erythrocyte carbonyl proteins (*n* = 1), heme oxygenase 1 (*n* = 1), nuclear factor erythroid 2-related factor-2 (*n* = 1), quinone oxidoreductase (NQO1) (*n* = 1), aldo-keto reductase family 1 member c1 (AKR1C1) (*n* = 1), 8-iso-prostaglandin F2α (1), ceruloplasmin (1), hs-CRP (*n* = 1), transferrin (*n* = 1), advanced oxidative protein products (AOPPs) (*n* = 1), protein carbonyl (PCO) (*n* = 1), GPx3 (*n* = 1), protein (P-SH) (*n* = 1), total nitrite (*n* = 1), non-protein thiol (NP-SH) (*n* = 1), total thiol (*n* = 1), non-protein thiol (NP-SH) (*n* = 1), P66Shc mRNA (*n* = 1), Drp1 mRNA (*n* = 1), protein ROS (*n* = 1), antioxidant enzymes and gene expression for mitochondrial function: ND2, TFAM, PGC1α, and NDUFB9 (*n* = 1) [21,22,23,24,25,26,27,28,29,30,31,32,33,34,35,36,37,38,39,40,41,42,43,44,45,46,47,48,49,50].

#### 3.4.3. Preterm Birth

Four articles about the role of oxidative stress in preterm birth were analyzed. All studies used a different set of OS biomarkers, none appeared in more than one of the studies. In total, 11 markers were measured, including 8-OHdG (*n* = 1), 8-isoprostane (*n* = 1), ROS (*n* = 1), GPx (*n* = 1), CAT (*n* = 1), NO (*n* = 1), O_2_^−^ (*n* = 1), peroxynitrite (OONO) (*n* = 1), hydroxyl radical (OH) (*n* = 1), 8-iso-prostaglandin F2α (*n* = 1) and prostaglandin F2α (*n* = 1) [51,52,53,54].

#### 3.4.4. General Pregnancy and Antenatal Care

Sixteen articles retrieved looked at pregnancy and general antenatal care. In total, 27 markers of OS were investigated in these studies. Parameters that were most frequently used were TAC (*n* = 7), GPx (*n* = 4), MDA (*n* = 4) and SOD (*n* = 3).

The rest of the markers were researched in either one or two studies: 8-isoprostane (*n* = 1), 8-OHdG (*n* = 2), total peroxide (*n* = 1), nitrotyrosine (*n* = 1), 8-iso-prostaglandin F2α (*n* = 2), 8-epiprostaglandin F2-α (*n* = 1), prostaglandin F2α (*n* = 1), thiol (*n* = 1), disulphide (*n* = 1), TOS (*n* = 1), TAS (*n* = 1), DNA damage in blood leukocytes (*n* = 1), CAT (*n* = 2), γ-glutamyl transferase (*n* = 1), hs-CRP (*n* = 1), GSH (*n* = 1), NO (*n* = 1), carbonyl proteins (*n* = 1), superoxide anion expressed as reduced nitroblue tetrazolium (*n* = 1), aldehyde dehydrogenase (*n* = 1), GST (*n* = 1), soluble fms-like tyrosine kinase-1 (*n* = 1), and placental growth factor (*n* = 1) [55,56,57,58,59,60,61,62,63,64,65,66,67,68,69,70].

### 3.5. Reproduction and Gynaecological Conditions

Twenty-three articles on reproduction and gynaecological conditions. Most conditions in which the association with oxidative stress was found are polycystic ovarian syndrome, endometriosis, and subfertility.

In total, 26 markers of oxidative stress were identified with particular emphasis on five markers: MDA (*n* = 11), TAC (*n* = 11), SOD (*n* = 10), ROS (*n* = 6), and GPx (*n* = 6).

The rest of the markers were: CAT (*n* = 4), GSH (*n* = 3), GR (*n* = 3), 8-Isoprostane (*n* = 3), 8-OHdG (*n* = 2), thiol (*n* = 2), LPO (*n* = 1), PON-1 (*n* = 1), advanced oxidation protein products (*n* = 1), TOC (*n* = 1), TOS (*n* = 1), TAA (*n* = 1), uric acid (*n* = 1), CRP (*n* = 1), IL-6 (*n* = 1), protein carbonyls (*n* = 1), TNF-α (*n* = 1), nitrates (*n* = 1), cortisol (*n* = 1), OSI (*n* = 1), and NO (*n* = 1) [71,72,73,74,75,76,77,78,79,80,81,82,83,84,85,86,87,88,89,90,91,92,93].

## 4. Discussion

We observed a huge diversity of markers used to describe oxidative stress. Almost every paper used a different set of markers, which made it challenging to compare and triangulate the results or perform a meta-analysis with cohesive conclusions. In the papers we reviewed, oxidative stress has been mentioned both as the exposure or the outcome. Certain papers described the use of antioxidants as a protective factor to prevent the aforementioned diseases. Therefore, there is a need for a cohesive and unified approach to be able to appropriately assess and define oxidative stress. Moreover, different abbreviations are used to describe the same parameter; in some cases, the abbreviation in the brackets stands for the laboratory technique rather than the acronym of the phrase.

Moreover, we discovered that different materials are being used to measure the markers of oxidative stress. For instance, in papers on polycystic ovarian syndrome we had markers retrieved from serum, blood, follicular fluid, or granulosa cells which all have different reference ranges and therefore it poses immense challenges of unifying and triangulating the results in order to make appropriate recommendations or conclusions.

Types of studies included in the final analysis varied in design. In many cases, the authors used different nomenclature to describe similar study designs, for example randomized controlled clinical trials and case-control studies often had similar methodology but authors used to describe them differently.

Additionally, in some studies we could observe a lack of disaggregation of the populations included in the study based on age and BMI—two known factors affecting oxidative status and stress. In light of the increasing number of non-communicable diseases deriving from obesity and its increased role in metabolic balance, it would be important to disaggregate specific populations in order to be able to avoid confounding results.

Finally, there is a clear need to differentiate between inflammation and oxidative stress markers. In many studies, the line between inflammatory and oxidative stress markers is not clearly stated and division is not well explained. For instance, C-reactive protein (CRP) is being used in many studies as a proxy for inflammation process; however, this might pose unnecessary confusion of comparing inflammation and oxidative stress markers as this division is not well explained, leading to potential interpretation errors.

Oxidative stress and antioxidants are becoming more popular in social media with regard to healthy diet culture as well as vitamin and other supplements intake. It is therefore extremely important to have unified definitions and markers of oxidative stress given that it might be the source of manipulation in the public discourse. Many pharmaceuticals and supplements are being advertised as antioxidants and gatekeeping them with the use of appropriate definitions and markers would allow validation and reliability, as well as replicability of the studies.

Finally, we would recommend creating a common, basic panel of oxidative stress markers that could be used in all studies on oxidative stress in obstetrics and gynaecology. This way, we could achieve reproducible results that could be further analyzed for oxidative stress to be better understood. The most commonly used markers of oxidative stress that we would recommend adding to the basic set are: reactive oxygen species (ROS)—as a direct marker of oxidative stress; 8-hydroxydeoxyguanosine (8-OHdG)—as a marker of DNA/RNA damage; and malondialdehyde (MDA)—as a marker of lipid peroxidation. Additionally, we would like to suggest adding two antioxidants parameters that are often used in studies—total antioxidant capacity (TAC) and gluthatione (GSH). Using the same basic set of oxidative stress markers would enable researchers to investigate and understand their actual clinical significance in order to create an even more adequate and reliable set of oxidative stress markers in the future. Moreover, we would like to recommend that the researchers use the basic set of proposed markers in order to standardize the studies on oxidative stress. However, the choice of additional markers should be made independently, depending on the studied disease and material.

## 5. Conclusions

There are no universal parameters assessing oxidative stress in human reproduction and pregnancy-related issues. In order to be able to appropriately derive conclusions, a unified set of parameters and definitions would be of use.

## Figures and Tables

**Figure 1 antioxidants-11-01477-f001:**
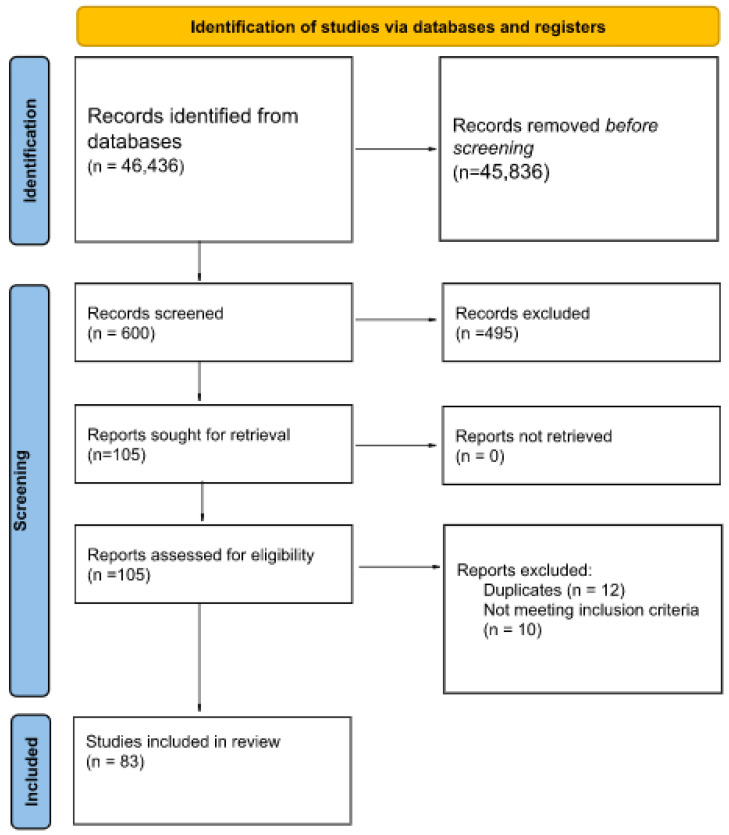
PRISMA diagram of the systematic literature review (*n*—number of records).

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
