# Peer review of "Markers of Oxidative Stress in Obstetrics and Gynaecology—A Systematic Literature Review"

_antioxidants, 2022, doi:10.3390/antiox11081477_

Round 1

Reviewer 1 Report

I appreciate the effort, but at this point it is just a list of measurements and does not contain much important information.

At this level, simply running a data mining program would provide more accurate information than human curation.

If researchers are doing the curation, then there needs to be some weighting information (what is important among the measurements).

Comments and suggested revisions have been added to the attached Word file using the ”Track changes".

Author Response

Dear Reviewer,

We would like to thank you for your review.

First of all, after conversations with the Editor and given suggestions from other reviewers the team has decided to narrow down the scope of the manuscript and focus on the markers of oxidative stress.

Next, we believe that by applying the rigorous methodology of systematic review we were able to curate a useful set of data that allowed us to identify and aim at unifying the markers of oxidative stress in obstetrics and gynaecology.

Finally, we appreciate your suggestions made in the manuscript, we certainly took them into consideration.

We hope that by incorporating those comments now the manuscript reads more easily and is much more improved and ready for the publication. 

Yours sincerely,

on behalf of the research team, 

Michalina Drejza

Reviewer 2 Report

The main objective of this literature review is to determine the different definitions of oxidative stress its markers in obstetrics and gynaecology. To this end, a large number of clinical studies relating the role of oxidative stress in reproductive diseases and pregnancy disorders are analysed.

The conclusion reached in this literature review is known from the beginning of the study: “there is no clear definition of oxidative stress and the markers and materials to quantify it are very diverse and different in each study”. Therefore, I believe that the authors should propose the best definition of oxidative stress in obstetrics and gynaecology, as well as the most appropriate markers and sample matrix to be used. Otherwise, the study would not provide any scientific novelty.

In addition, authors should consider including in the table the type of clinical study conducted and the quality indices of the journal in which it was published.

As positive points of the work it is necessary to indicate the following:

The title does accurately reflect the content of the article.

The materials and methods are well explained and reproducible.

The language used is clear and fits the scientific style.

The bibliographic references are adequate and are limited to the last 10 years.

Author Response

Dear Reviewer,

We would like to thank you for your extremely helpful review.

First of all, thanks to your suggestion we decided to narrow down the scope of the manuscript and focus on the markers of oxidative stress. Therefore, we have decided to edit the title in order for it to more appropriately reflect the methodology and results of our study.

Moreover, the Team added a column in the main table to include the types of the studies included in systematic review. We felt that the quality indices were not necessary as quality appraisal was incorporated into the inclusion criteria in our methodology. 

We hope that by incorporating those suggestions now the manuscript reads more easily and is much more improved and ready for the publication. 

Yours sincerely,

on behalf of the research team, 

Michalina Drejza

Reviewer 3 Report

Dear Authors,

I have read the praiseworthy manuscript titled Definition of oxidative stress and its markers in obstetrics and gynaecology - scoping literature review, in which you have attempted to lay out an analysis as to oxidaive stress markers by combing through relevant sources and defining a degree of objectivity on a highly meaningful ob/gyn issue.

I believe the article's chief strength resides in its relevance and potential interest to the scientific community.

As a matter of fact, oxidative imbalances are thought to have the potential to negatively affect fetal development over the pregnancy, and although no conclusive evidence has as yet been produced, such associations have been debated by researchers and clinicians. I believe the article has been competently assembled and relies on sound methodology, as far as I was able to determine.

Reactive oxygen species (ROS) and reactive nitrogen species (RNS) are both known to play a meaningful role as secondary messengers in many intracellular signalling cascades. The authors have described and discussed such processes only in passing (page 2). I feel that the discussion ought to be expanded for the sake of thoroughness, by including a more comprehensive description of the critical effects of intracellular signaling, in light of ROS and RNS, on the pathological processes affecting the pregnant woman. The underlying mechanisms of oxidative stress in pregnancy has also been linked to pregnancy-related clinical conditions such as preeclampsia. Moreover, data show how N-acetyl-cysteine (a by-product of glutathione which impacts glutathione maintenance and metabolism) may affect oxidative stress in preeclamptic pregnant women. The results indicated that supplementation with N-acetyl-cysteine improved liver and kidney function, decreased blood pressure, decreased proteinuria and ameliorated the severity of oxidative stress in preeclampsia. Again, the authors make no mention of this and cover preeclampsia only in passing, but fail to cite relevant sources and further elaborate on such linkages by adding an analysis of their own. Reviewing scientific findings should be viewed as a starting point to construct an original analysis based on one's own expertise, and I feel the authors have left that part a bit underdeveloped.

Overall, the article needs further proofreading by a native speaker of English. It is quite well written, but some occasional instances of clumsy grammar and less than ideal vocabulary choices detract a bit from its clarity and readability. The figures are acceptable and clearly crafted enough, but Figure 1 is a bit on the blurred side. Please improve it.

Looking forward to reviewing an improved version of this compelling review article.

Best,

Author Response

Dear Reviewer,

We would like to thank you for your very thorough and supportive review.

We have taken your comments into consideration and expanded the introduction with a more in-depth description with regards to links of ROS/RNS and pregnancy and links of oxidative stress and preeclampsia. We have also linked appropriate references. 

Moreover, the team has assured additional proofreading of the manuscript and edited the figure that weren’t showing properly as advised.

We hope that by incorporating those comments now the manuscript reads more easily and is much more improved and ready for the publication. 

Yours sincerely,

on behalf of the research team, 

Michalina Drejza

Round 2

Reviewer 1 Report

No more comments.

Author Response

Dear Reviewer,

Thank you very much for your positive review. We have worked relentlessly to improve our manuscript and it would not be possible without your constant support.

Best wishes,

Michalina Drejza

Reviewer 2 Report

Dear Authors

I consider that the manuscript has been significantly improved after the work of synthesis and organisation of the results.

Reducing the scope of the study and changing the title was a good idea.

However, I still believe that in order to publish the article the authors should propose the best definition of oxidative stress, as well as the most appropriate markers and sample matrix to be used.

Sincerely

Author Response

Dear Reviewer,

We would like to thank you for your review and recommendation to formulate a recommendation for a basic set of oxidative stress parameters. After internal discussions we decided to add our proposed set in order to unify parameters used to describe oxidative stress in studies that pertains to obstetrics and gynaecology. This can be found in the discussion section of the article.

Best wishes,

Michalina Drejza

Reviewer 3 Report

Dear Authors,

I believe that overall, you have succeeded in improving your manuscript to a substantial degree. The article is competently assembled and has a coherent structure, also relying on an effective table conveying the studies' characteristics in great detail.

I believe the improvements made by the authors warrant approval for publication.

Sincerely,

Author Response

(The authors gave the same response as above.)
